# The Territory of Valle del Jerte-La Vera and Its Tourist Development (Extremadura, SW Spain)

Nerea Ríos Rodríguez, Gema Cárdenas Alonso, Ana Nieto Masot * and Felipe Leco Berrocal

Department of Art and Territorial Sciences, University of Extremadura, 10003 Cáceres, Spain
* Correspondence: ananieto@unex.es

**Abstract:** At the end of the 20th century, tourism was positioned as an activity capable of diversifying and reactivating the economies of rural European areas, which were experiencing problems of demographic regression and a high rate of ageing. Subsequently, with the emergence and promotion of new models of tourism consumption, the provision of rural tourism facilities has increased, as is the case in the north of Extremadura. This study analyzes, through the use of a descriptive and analytical method, the distribution of the demographic, socioeconomic and heritage variables existing in the tourist territory of the Valle del Jerte-La Vera region in order to interrelate them with the tourist supply and demand in this area. The results allow us to observe that Valle del Jerte-La Vera is promoting tourist activities, with the promotion and implementation of better lines of action for the reception of travelers, in such a way that they favor the increase in the economic income, and these factors are capable to maintain the existing population, thus facilitating the development of rural areas.

**Keywords:** rural tourism; travelers; overnight stays; Extremadura

## 1. Introduction

In the middle of the 20th century, tourism developed significantly, and it was consolidated as a mass phenomenon [1–3]. In Spain, the tourism sector has been evolving over the years to adapt to the new needs and demands of travelers and began to play an important role in the country's economy from the 1960s onwards, with the "sun and beach" tourism model, given its territorial conditions. Subsequently, in the late 1980s and early 1990s, this model declined and deteriorated [4] due to the uncontrolled increase in the number of tourists concentrated in the same tourist destination. This caused the problem of overcrowding and triggered a crisis in the coastal areas due to a change in the preferences of travelers, who began to opt for short-stay trips to destinations close to cities and in areas that were not very developed. In recent decades, technology, the great variety of accommodation and the discovery of new places have played an important role in this process. All of this has favored inland tourism, which is developed away from the coasts, and its offer is oriented towards the enhancement of heritage, cultural and/or natural resources [5–7]. In this way, there has been a commitment to the development of new tourist destinations, mostly in rural areas, in which little transformation of the territory and a wide range of natural and cultural enclaves predominate [8,9], promoting quality tourism, the seasonal diversification of the supply and the extension of the latter to the interior of the peninsula [10].

Rural areas have come to diversify their uses, and, in many of them, the predominant activity has become tourism based on leisure activities that take advantage of the resources and products of each area, generating their development. This has given rise to the appearance of different types of tourism (nature tourism, green tourism, ecotourism, agrotourism, adventure tourism, cultural tourism, alternative tourism, etc.) and the inclusion of different activities, such as gastronomy, horse riding, hunting, fishing, other sports, cultural and

historical visits, etc. [11]. Thus, in rural areas, tourism has become a strategic sector for their wealth, establishing itself as an activity that can complement other traditional activities, such as agriculture, livestock farming, crafts or small industry. Its contribution to the gross domestic product (GDP) is important, as well as to the creation of employment and to the capacity to dynamize visitor flows throughout the year, favoring the seasonal diversification of the tourism demand [12], especially in rural areas affected by depopulation and the growing crisis in their productive model. Tourism is contributing to the positive economic growth and the revitalization of disadvantaged or isolated areas, as well as to the maintenance or growth of their populations [13–15].

In Spain, one of the tourist destinations that has benefited the most from the changes in tourist habits, has been the region of Extremadura [16]. Since 2000, the number of travelers in the region has increased by 35% (the Spanish average is 41.29%), higher than destinations, such as Castilla y León (32.92%) or the Canaries (24.97%), and, consequently, it has experienced a 40% increase in overnight stays (much higher than the Spanish average of 25%) and a 54.9% increase in the number of establishments. However, the contribution of tourism to the GDP is still below the Spanish average (12.4%), although it is the economic sector whose contribution to the GDP is growing the most in the region [17].

In recent years, the new desires of travelers, and their search for experiences, rather than stays, have highlighted the need to increase investment towards a qualitative improvement in tourism [18]. In fact, in recent decades, the European Union (EU) has promoted a series of actions and economic policies aimed at developing tourism and agrotourism in rural areas through national and European policies with structural funds or specific initiatives, such as INTERREG or LEADER [15,19–22], for its transformation into a complementary activity to the agricultural income in areas affected by depopulation [23,24]. In this sense, one of the areas in Extremadura that is receiving the most attention is the Valle del Jerte-La Vera tourist area, the study area of this paper, as it has rich natural and cultural resources that can be exploited for tourism and is one of the areas with the highest demand for travelers in the region.

In the following, taking into account the above considerations, the literature review section is presented. Following this, the methodology used for the development of the research is established, as well as the delimitation of the study area. Section 3 shows the results obtained. These are discussed in Section 4, together with the conclusions extracted.

*Literature Review*

In general terms, the typology of tourism in rural areas has many common characteristics, as established by several authors [14,25–28]. It is a type of tourism that is developed in rural areas with little massification, with a quality cultural and natural offer, linked to the rural space [29], contributing as an activity for the local development of rural areas, thus maintaining the existing population, in order to avoid a rural exodus [30,31]. Due to the space in which this tourism takes place, it can be complemented by other types of tourism [32], such as ecotourism or nature tourism, mountain tourism, adventure tourism, agrotourism, cultural and historical tourism, religious, astronomical, ornithological, hunting, gastronomic, wine tourism, etc.

Taking into account these types of tourism, many of them have been developed in the region of Extremadura, with the mainly highlighted cultural tourism, nature tourism and gastronomic tourism [33]. Nowadays, there is a greater demand for other activities, such as those linked to agricultural activities, which is why agrotourism has been developed in many areas of the region of Extremadura [34,35].

Several authors [22,36] have analyzed tourism activities in this area, mainly in Valle del Jerte, providing information on the profile and interests of visitors, as the case of Millán [37], who has analyzed the rural travelers, stating that "rural tourism requires integration with its environment and its people, because rural tourists are looking for more than just lodging, they are looking to soak up the colors, flavors and smells of the rural environment. The balance between the ecological, socioeconomic and cultural systems of the area guarantees

the offer of an attractive product, when combined with adequate lodging infrastructures and specialized and personalized attention".

Engelmo et al. [38] have analyzed the aid from European funds, such as LEADER, and affirmed that this territory benefits from its proximity to the capital of the country and presents a widely developed offer promoted by all local agents, based around its emblematic agricultural production, the cherry tree, and the natural heritage with figures of protection, such as the Garganta de los Infiernos Natural Reserve.

From the point of view of tourism marketing and promotion initiatives, studies of Mediano and Vicente [25], Fyall et al. [39] and Di Clemente et al. [40] highlight, where they state that a good promotion and a tourism marketing make the area known, and this will increase the attractiveness of the area [25,39–41]. Other authors [42,43] analyze tourism marketing activities, taking into account the characteristics of the demand, since one of the objectives of the marketing is to meet the needs of travelers. Linked to the tourism promotion, in recent years, Valle del Jerte-La Vera has been established as a smart tourist destination, in order to develop a greater digitization, sustainability, accessibility, quality or innovation that generates an increase in tourism demand and also has an impact on improving the perception of the destination and the tourist experience.

In view of the above, the aim of this work is to analyze the tourism sector in the tourist territory of Valle del Jerte-La Vera, to obtain information about tourism activities from 2018 to the present day and its relationship with the socioeconomic, demographic and heritage variables, which will allow us to identify the resources this territory possesses and whether these are a key element for the development of its tourism activities.

## 2. Materials and Methods

At the methodological level, this study gathers the relevant information related to the tourist, demographic, socioeconomic and heritage activities of the tourist territory Valle del Jerte-La Vera, alphanumerically and cartographically. This information is analyzed through the descriptive method [44–46], which allows us to perform the territorial characterization of the area, in order to determine whether its resources serve as a focus of attraction for travelers. As Gómez [47] states, "descriptive studies seek to specify the properties, characteristics and important aspects of the phenomenon under analysis". It should be noted that the use of this methodology was adopted, in order not only to collect information on its characteristics, but also to be able to use these results in subsequent studies.

### 2.1. Study Area

The study area of this research (Figure 1) is the tourist territory Valle del Jerte-La Vera, located in Western Spain, specifically, in the Autonomous Community of Extremadura. Within the Extremadura region, this tourist territory borders the tourist territories of Valle del Ambroz-Tierras de Granadilla to the northwest, Plasencia to the southwest and the Reserva de la Biosfera de Monfragüe to the south, while the rest of its perimeter borders the provinces of Ávila and Toledo.

The tourist territory of Valle del Jerte-La Vera covers an area of 1258.61 km$^2$ and has a perimeter of 189 km. It also includes a total of 30 municipalities, of which 11 belong to the Valle del Jerte region and 19 to the La Vera region. This tourist territory, together with 14 others, brings together the 388 municipalities that constitute Extremadura and represent the territorial demarcation of the tourist territories established by the Directorate General of Tourism of the Junta de Extremadura, which is used in this study as the spatial scope of action for the design of the different tourism policies in the region [48].

The Valle del Jerte-La Vera territory has a total of 34,456 inhabitants, making it the third least populated tourist territory, although it has a population density of 27.4 inhabitants/km$^2$, above the regional average (25.5 inhabitants/km$^2$). In terms of variables, such as the ageing index and the youth index, the average for this territory is 328.3% (24.9% regional) and 10.4% (113.2% regional), respectively, which highlights the advanced ageing of the population, a highly characteristic and concerning aspect of Extremadura's mountain areas.

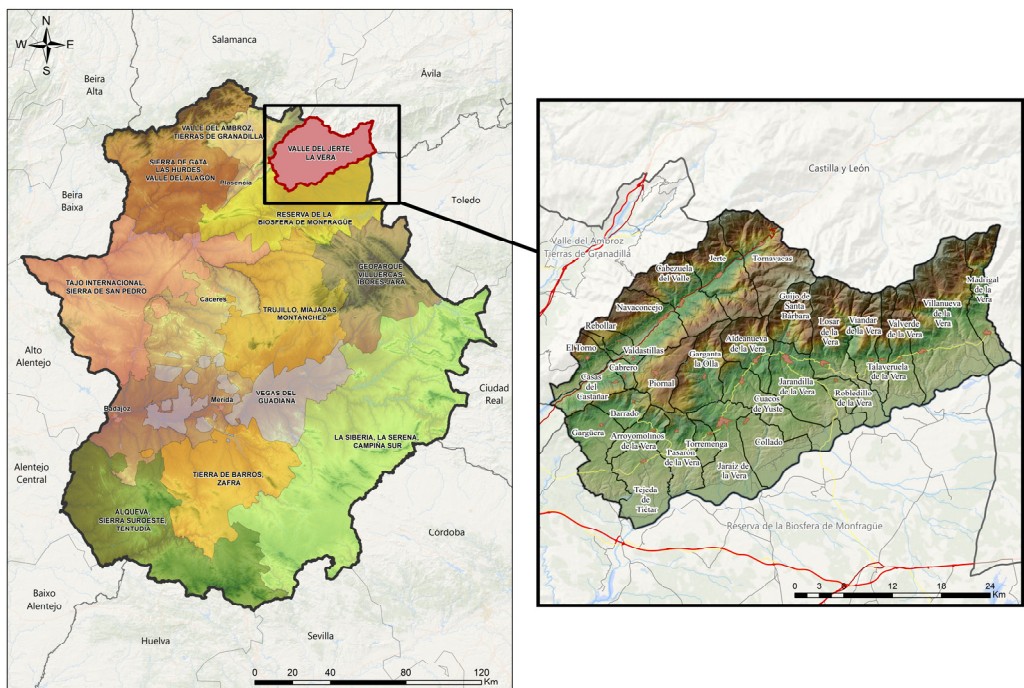

**Figure 1.** Map showing the location of the tourist territory of Valle del Jerte-La Vera.

It should be noted that access to the area is by land, as the only existing means of communication in the area is roads. The roads that provide access to the area are divided into various levels, including national roads (N-110), various regional roads (EX-119, EX-203, EX-213, EX-392, etc.), provincial roads and other types of regional roads. Tourism and means of transport, together with their infrastructures, are intrinsically linked, since, without travel, there is no journey [49]. In this way, the means of transport and their means of communication are some of the driving forces that allow tourism to function.

With regard to the physical environment, it is a predominantly mountainous territory, with markedly differentiating characteristics. Due to its natural and scenic values and the biodiversity of its surroundings, this territory contains protected natural areas (45.93% of the protected hectares of the tourist territory). In terms of climatic characteristics, this area is conditioned by its relief and orography, given that the mountain ranges prevent the incidence of northerly winds and favor those from the Atlantic Ocean, which generates cold winters and mild summers. All of these characteristics give the territory a high level of landscape, climatic and natural richness, which attracts tourists.

### 2.2. Databases

The most important point in the methodological process (Figure 2) of this research is the construction of the cartographic and alphanumeric databases, which were developed after the review of the documentary sources and the selection of the data necessary for the analysis.

Firstly, a cartographic database was created from the polygonal cartography layer of the municipalities obtained from the National Cartographic Base 1:200,000 (BCN200) of Spain's National Geographical Institute. This layer was used to obtain, as tabular information, the 30 municipalities belonging to the area under study and corresponding to the tourist territory of Valle del Jerte-La Vera delimited by the Tourism Observatory of Extremadura and the Junta de Extremadura (Figure 3).

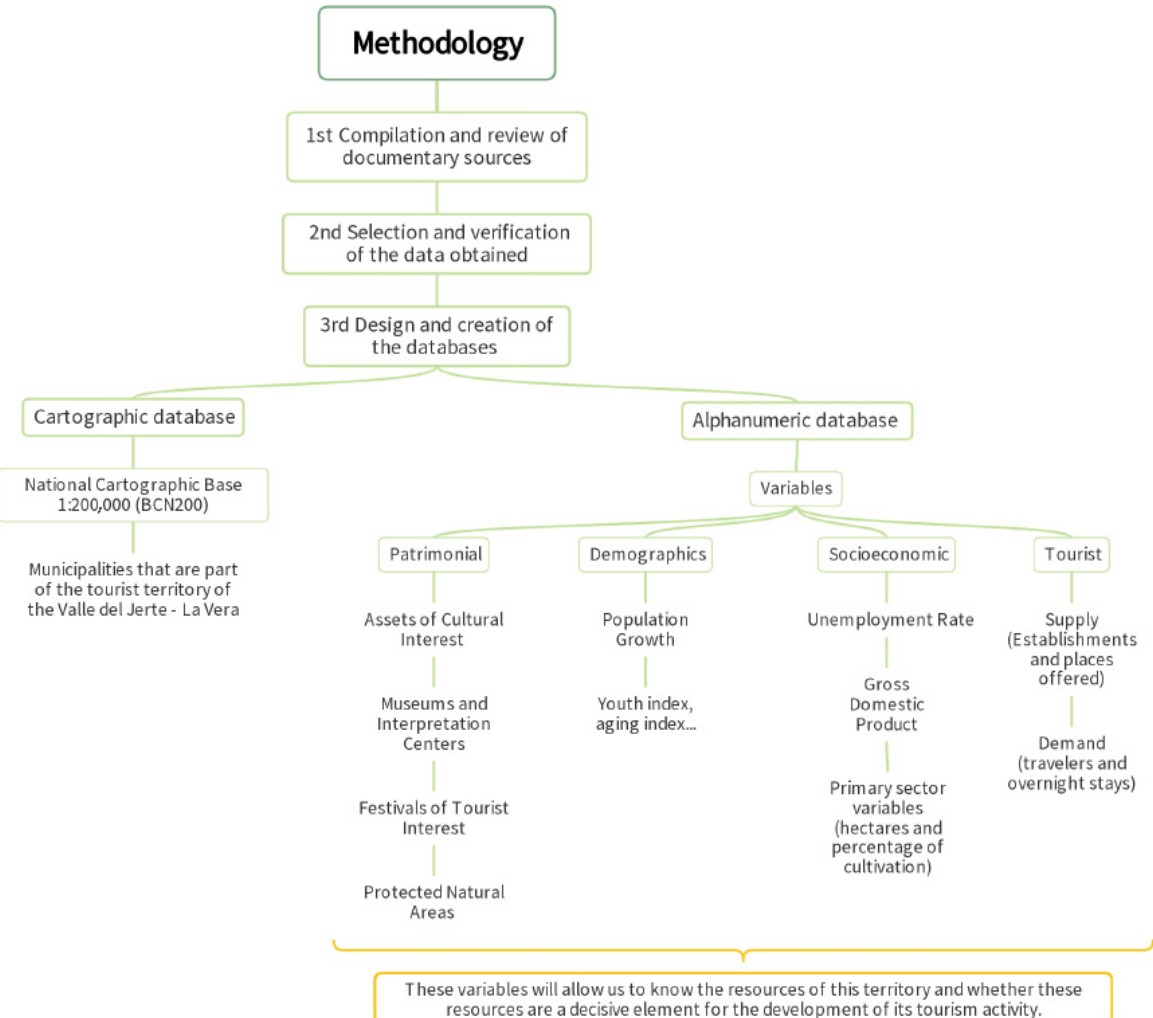

**Figure 2.** Methodological scheme.

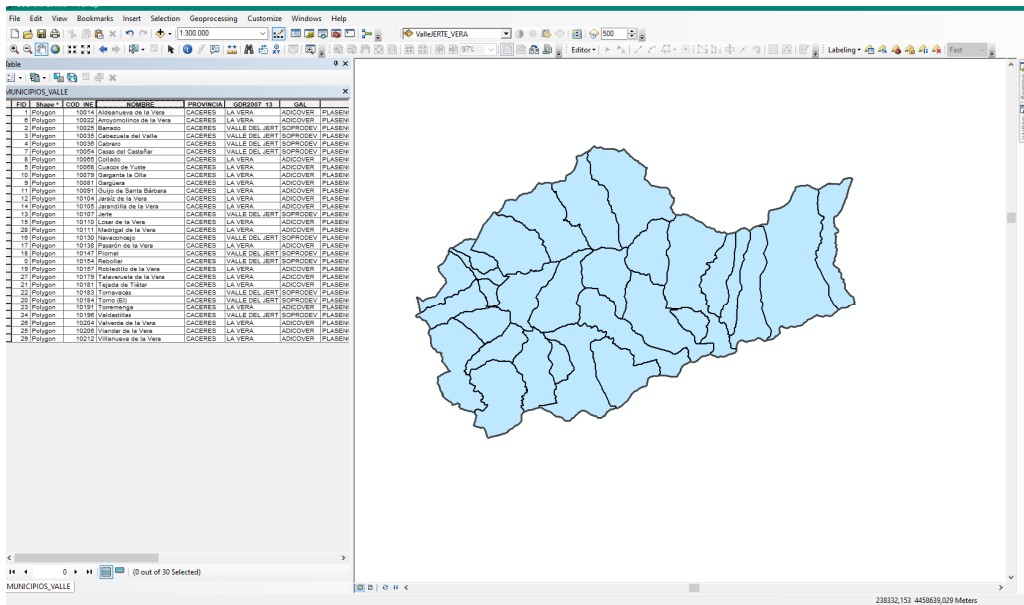

**Figure 3.** Polygonal layer of the municipalities of the tourist territory with alphanumeric information.

Then, alphanumeric databases were designed and created, according to heritage, demographic, socioeconomic and tourism variables.

The information on heritage resources, both cultural and natural, was collected. These resources were as follows.

- Assets of Cultural Interest. The database of immovable assets registered in the Register of Assets of Cultural Interest of the Ministry of Culture and Sport of the Spanish government was consulted, and the information was completed with that obtained from the Extremadura Observatory of Culture.
- Museums and Interpretation Centers. Information was obtained from the Extremadura Observatory of Culture and, in the case of the Interpretation Centers, those established on the Extremadura Tourism website were updated.
- Festivals of Tourist Interests. Data were downloaded and updated from the Spatial Data Infrastructure of Extremadura and the Extremadura Observatory of Culture, as these festivals are an element of attraction for tourists in the region.
- Protected Areas. Information on the protected natural areas of Extremadura, European Ecological Network NATURA 2000, was obtained from the Extremambiente website of the Junta de Extremadura.

With regard to the demographic variables, the evolution of the population of each of the municipalities that constitute the tourist territory was analyzed. Furthermore, the ageing index, the youth index and the migratory balance were calculated in order to ascertain the variation in the population of this area.

Then, the socioeconomic variables and the gross domestic product (GDP) for 2019 were analyzed, and the unemployment rate (July 2021) and other variables, related to agriculture (hectares and percentage of cultivation, etc.) in the area, relating to 2019, were downloaded from the Socio-Economic Atlas of 2021 of the Extremadura Statistics Institute.

Finally, with respect to the tourism sector, the tourism supply and demand for the years 2018, 2019, 2020 and 2021 were studied. On the one hand, we studied the data on the supply of tourist infrastructures, the number of establishments and the existing vacancies per establishment in each municipality of the tourist territory, obtaining all of the establishments from the Register of Tourist Companies of the Directorate General of Tourism of the Regional Government of Extremadura. Taking these into account, those establishments that completed the occupancy surveys for each of the years were analyzed and grouped into three types of accommodation: hotels (hotels, hostels and guesthouses), non-hotels (tourist flats, hostels and campsites) and rural (rural flats, rural hotels and rural houses and other buildings dedicated to the agritourism sector, located in rural areas). On the other hand, in the case of tourist demand, the variables of total, national and international travelers and overnight stays were used, also provided by the Directorate General of Tourism of the Junta de Extremadura and by the Tourism Observatory of Extremadura. The data on travelers and overnight stays were structured according to the type of tourist accommodation, by month and corresponding year. It should be noted that tourist demand had to be analyzed for Valle del Jerte-La Vera as a whole, since the values at the municipal level were estimated, in order to obtain the data for tourist territories, regional territories, etc.

All of these variables (Figure 2) were linked to the cartographic database, in order to understand in greater detail, the reality of the territory of study.

## 3. Results

### 3.1. Territorial Characteristics

Over the last century, according to data from the National Institute of Statistics of Spain (NIS), the demographic evolution of the Valle del Jerte-La Vera territory, on par with the rest of Extremadura, has not been positive. Figure 4 shows how the total population of the territory has lost more than 3600 inhabitants over the last 20 years, with a slight stagnation in 2010 and 2011.

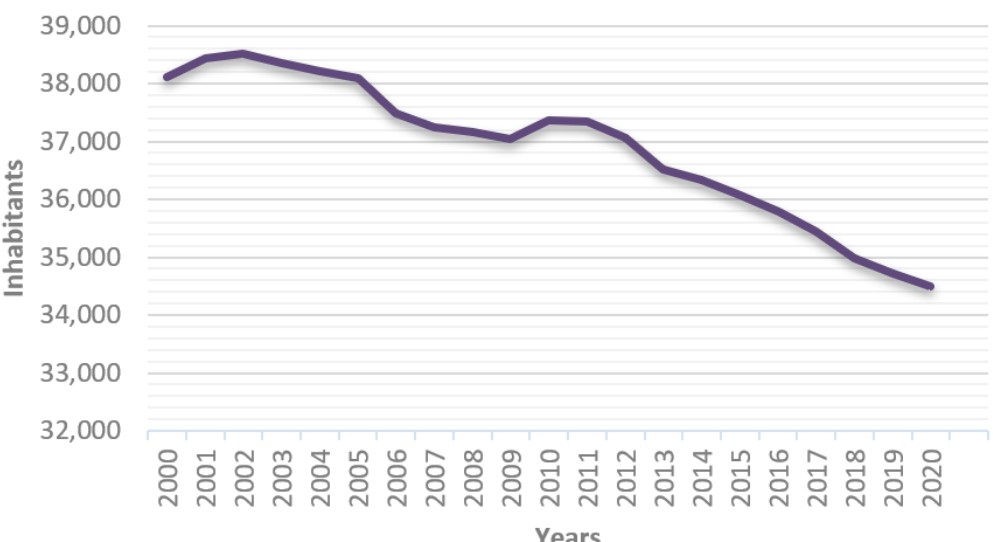

**Figure 4.** Evolution of the population (2000–2020) in the tourist territory of Valle del Jerte-La Vera.

Considering the population figures by municipality, it is worth mentioning that, in the last decade, practically all of the municipalities in the study area experienced a reduction in their population, reaching values of more than −23% in the case of Valverde de la Vera. Moreover, Collado de la Vera and Gargüera stand out, with their populations increasing by 15.5% and 44.7%, respectively, both with figures of between 150 and 200 inhabitants. Equally, there are municipalities which, in the last 20 years, have increased their numbers of inhabitants, such as Navaconcejo (2.9%), Torremenga (5.8%) and Villanueva de la Vera (6.9%). It is noteworthy that, at present, these municipalities do not have more than 2200 inhabitants.

According to the distribution of the population by municipality, Jaraíz de la Vera contains the largest number of inhabitants (6499 inhabitants), followed, with less than half, by Jarandilla de la Vera (2843), Losar de la Vera (2658) and Cabezuela del Valle (2141), among others. Thus, the municipalities in this area are characterized by a low population concentration, with 19 municipalities below the 1000 population threshold, of which only seven have more than 500 inhabitants. The population density is 27.38 inhabitants/km$^2$, which is higher than the regional average (25.45 inhabitants/km$^2$). There are marked contrasts in the territorial distribution of the population, with 12 municipalities showing values well above the regional average, including Jaraíz de la Vera (103.88 inhabitants/km$^2$), Aldeanueva de la Vera (54.09 inhabitants/km$^2$), Cabrero (48.72 inhabitants/km$^2$) and Torremenga (48.27 inhabitants/km$^2$). These high values indicate an intensity of population that is due more to the small surface area of their municipalities than to the volume of the resident population. Moreover, in terms of variables, such as the ageing index and the youth index, the average for the tourist territory is 328.3% (113.2% for the region) and 10.4% (24.9% for the region), respectively, which highlights the advanced ageing of the population, much more significant in the mountain areas. Moreover, in relation to this, the birth and death rates reflect a low birth rate, which almost reaches 4‰, compared with almost 7‰ in the region as a whole, and a high mortality rate (18‰—six percentage points higher than the regional mortality rate, which is 12.3‰). It is necessary to take into account the migratory balance, as this area's population increased by 275 inhabitants in 2020, an aspect that may be due to the increase in the population registered in the municipality as a result of the pandemic. Thus, due to the population density, advanced ageing and the small number of inhabitants living in these municipalities, this territory has a markedly rural character.

From the socioeconomic point of view, in recent years, the activity in Valle del Jerte-La Vera has increased, showing a great difference by sex and with the percentage of active women being much lower than that of men. In terms of the GDP, corresponding to the

year 2019, most of the municipalities had a GDP of more than EUR 7000/inhabitant, with Robledillo de la Vera standing out, with more than EUR 42,000/inhabitant, as well as Valdastillas, with values close to EUR 31,000/inhabitant. At the bottom of the ranking are Collado de la Vera (EUR 6429/inhabitant), Rebollar (EUR 6389/inhabitant) and Gargüera (EUR 5932/inhabitant). Looking at the unemployment rate, only six municipalities (Gargüera, Valdastillas, Robledillo de la Vera, Cabrero, Valverde de la Vera and Talaveruela de la Vera) exceed the regional rate (13.9%). This percentage is higher, considering the rate for the study area (11.5%), which is exceeded or equaled by 11 of their 30 municipalities.

In terms of sectors of activity, the primary sector, specifically agriculture, still plays an important role for the employed population. According to the Socio-Economic Atlas of Extremadura for the year 2021, whose data correspond to the year 2019, a total of 16,168 cultivated hectares are concentrated in this area, of which 10,691 are rainfed and 5477 are irrigated (Figure 5). A more detailed analysis of their distribution by municipality shows that Jaraíz de la Vera, Navaconcejo, Losar de la Vera and Cuacos de Yuste have more than 1000 cultivated hectares, while others, such as Robledillo de la Vera, Gargüera, Viandar de la Vera and Guijo de Santa Bárbara, do not reach 200 hectares. The distribution of dry and irrigated hectares is closely related to the presence of watercourses in the municipalities bordering the territory of the Reserva de la Biosfera de Monfragüe, which results in a greater presence of irrigated cultivated hectares in the southeast of the territory. The most representative crops in Valle del Jerte-La Vera are fruit trees (7864 ha), mainly cherry trees, followed by industrial crops (3565 ha), with pepper for paprika as the main crop, and olive groves (3308 ha).

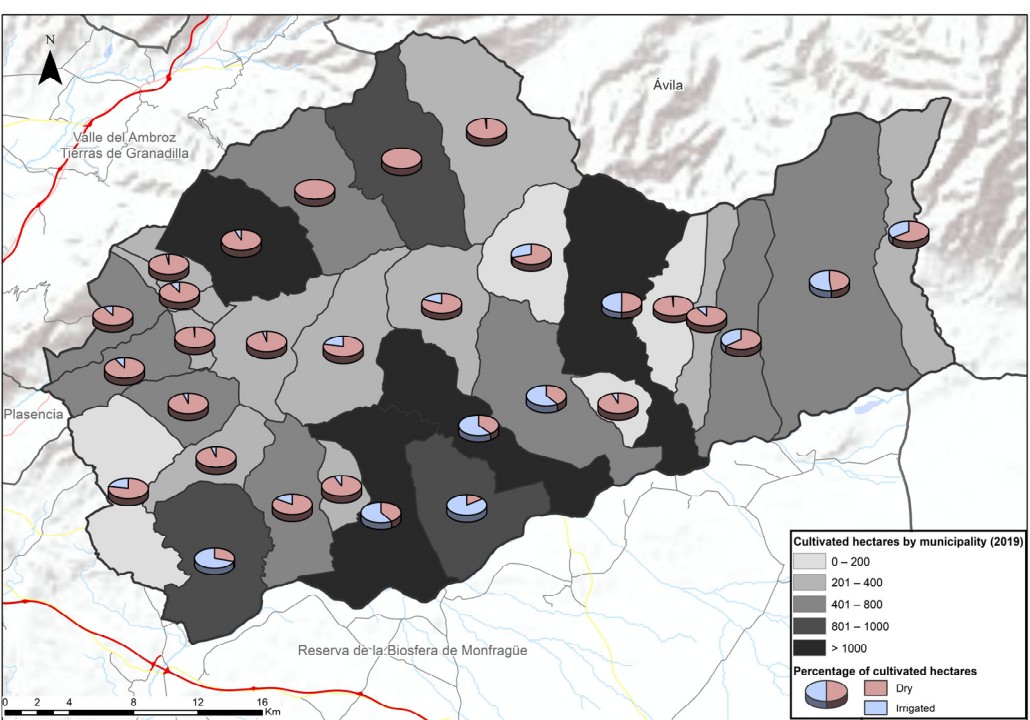

**Figure 5.** Map of the cultivated hectares and their typology by municipality.

Nowadays, the percentage of the population employed in the service sector has increased, largely due to tourism and, in many cases, to the implementation of economic diversification projects, such as those subsidized through the EAFRD and the LEADER method. Linked to this, their development strategy is closely related to endogenous products, as is the case for cherries in Valle del Jerte and paprika in La Vera.

In terms of heritage resources, numerous civilizations have passed through Extremadura throughout history, with their cultural legacy extending to the present day. This, together with the great natural and scenic diversity of the environment, means that there is high

heritage value. This is why the territory has a large number of cultural resources, some protected and others to be protected, with a large extension and quantity of natural resources spread throughout the territory.

The following map (Figure 6) shows which municipalities in Valle del Jerte-La Vera have the highest number of cultural heritage elements. The area has a total of 19 assets of cultural interest, concentrated in municipalities, such as Cuacos de Yuste and Pasarón de la Vera, while 17 municipalities have none. In addition, the area is home to 15 museums, with Cabezuela del Valle and Garganta la Olla with several museums each, while 17 municipalities have no museums at all. The number of interpretation centers is smaller, with only six in Valle del Jerte-La Vera. Taking these three heritage-related elements into account, 11 of the 30 municipalities do not have any of these typologies.

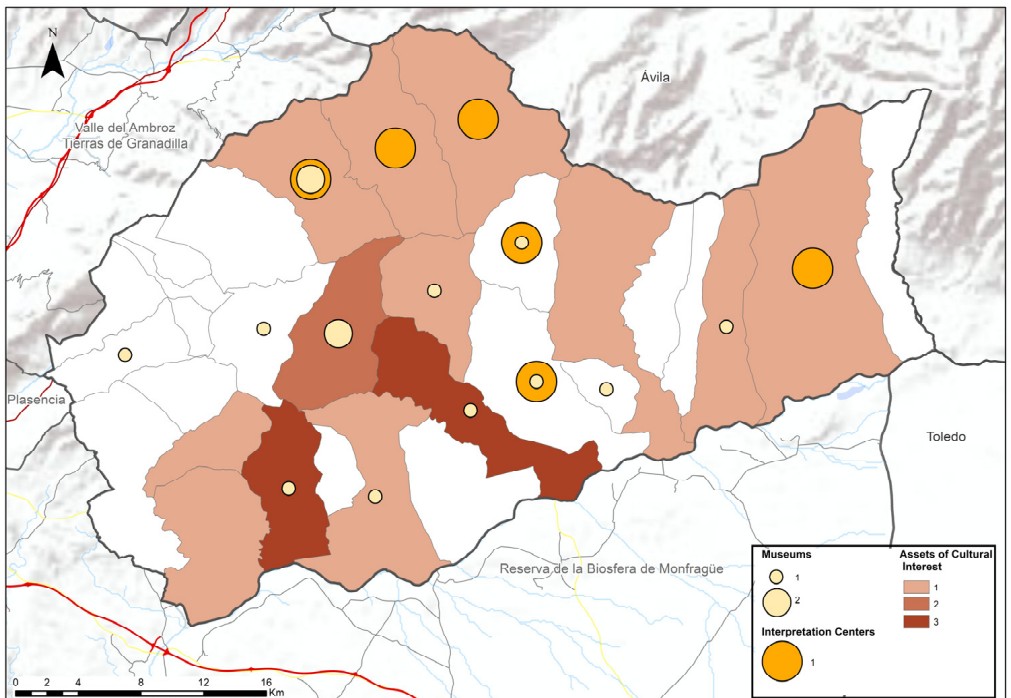

**Figure 6.** Map of the assets of cultural interest, museums and interpretation centers.

In Valle del Jerte-La Vera, there are several festivals of tourist interest, including the Cherry Blossom Festival, of national tourist interest, which covers the entire area of Valle del Jerte. This festival is one of the main tourist attractions, as this valley has more than two million cherry trees, which, when in bloom, give rise to a unique spectacle and landscape. The festival also tries to reflect the life of the whole region, of a land with an agricultural tradition and a star product, the cherry. To this end, all types of activities are organized to showcase the region's culture, gastronomy, traditions and way of life.

Furthermore, of national tourist interest is the Jarramplas festival, which is held in the municipality of Piornal. Of regional interest are Los Escobazos, in Jarandilla de la Vera, Los Empalaos, in Valverde de la Vera, and El Peropalo, in Villanueva de la Vera. All of these festivals are characterized by representing special moments in the lives of their populations, by reproducing manifestations related to religion, rites, fights, etc., in such a way as to provide a better understanding of the customs of these municipalities.

In terms of natural heritage, the Network of Protected Areas of Extremadura is composed of the Network of Protected Natural Spaces of Extremadura, under 10 protective figures (natural park, nature reserve, natural monument, protected landscape, area of regional interest, ecological and biodiversity corridor, Periurban Park for conservation and leisure, site of scientific interest, singular tree and ecocultural corridor) and the European Ecological Network Natura 2000, with special protection areas for birds (SPAs) and sites of

community interest (SCIs), singular tree and ecocultural corridors, the latter called special areas of conservation (SACs) following the publication of the management plans in Decree 110/2015, of 19 May, which regulates the European Ecological Network Natura 2000 in Extremadura [50]. These protected areas occupy more than 1,250,000 ha in our region, of which 57,806 ha are located in the tourist territory of Valle del Jerte-La Vera. The following map (Figure 7) shows a spatial representation of the Network of Protected Natural Spaces in the study area.

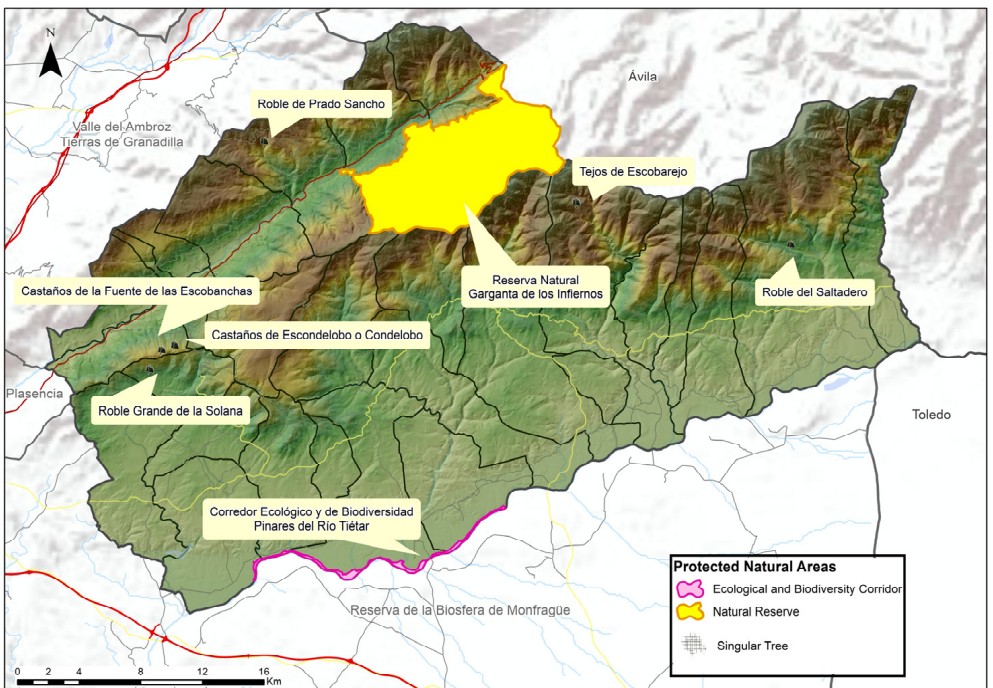

**Figure 7.** Map of the Network of Protected Natural Areas.

According to the European Ecological Network NATURA 2000 (Figure 8), there are two SPAs in Valle del Jerte-La Vera. These are the Lesser Kestrel Colony SPA in Jaraíz de la Vera, which covers an area of 33.17 ha and is characterized as an urban SPA, and the Río y Pinares del Tiétar SPA, which extends through numerous municipalities in the area. Moreover, the SACs are areas where the necessary conservation measures are applied for the maintenance or re-establishment of natural habitats and/or species populations. The Valle del Jerte-La Vera territory has four SACs: Sierra de Gredos and Valle del Jerte (69,528.61 ha), Monasterio de Yuste (13.81 ha), River Tiétar (4321.03 ha) and Rivers Alagón and Jerte (3131.70 ha).

The Valle del Jerte-La Vera area has a large number of water resources of considerable importance, among which the Garganta de los Infiernos stands out [29,51]. This is home to more than 10 natural pools, characterized by their shape (small natural pits called "marmitas"), due to the erosion of water on the granite rock of the area over millions of years. Other gorges in the area are the Garganta de las Monjas, near Cabezuela del Valle, the Garganta de las Nogaledas, near Navaconcejo, the Garganta de Alardos, in Madrigal de la Vera and the Garganta Bohonal, between Piornal and Valdastillas. There are also other bathing areas, such as Las Pilatillas, in Garganta la Olla and El Lago, in Jaraíz de la Vera.

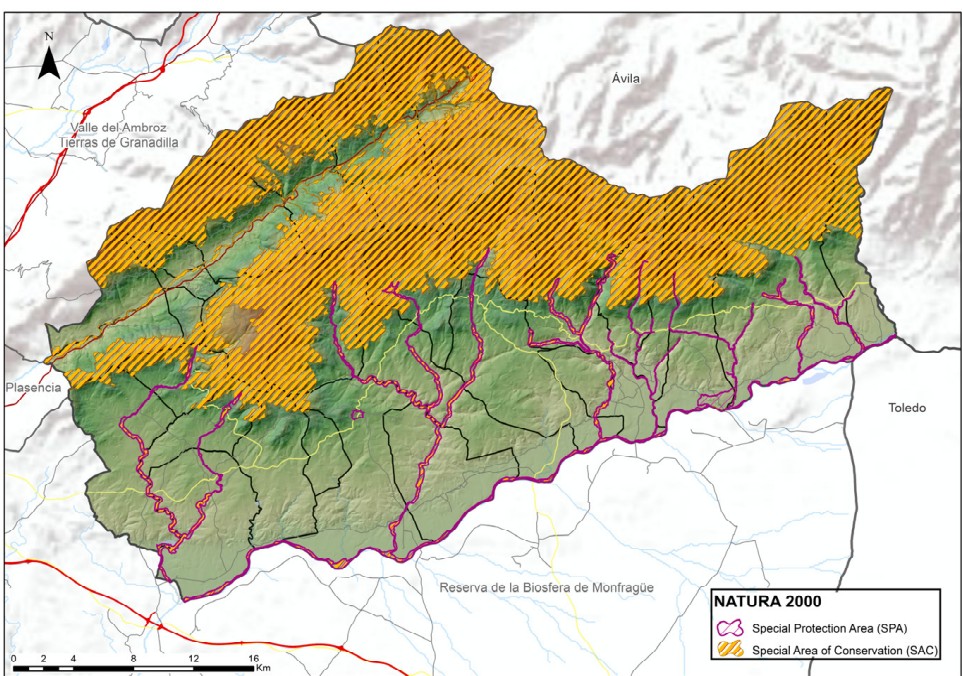

**Figure 8.** Map of the European Ecological Network Natura 2000.

### 3.2. Tourism Characteristics

Over the last few years, Valle del Jerte-La Vera has consolidated its position as the tourist territory in the Extremadura region with the highest number of accommodation establishments and vacancies offered. Moreover, and taking into account the type of establishment owner, as will be explained below, in this area, there is a large number of female promoters, especially in rural and non-hotel accommodations. Analyzing its tourist supply and demand, taking into account the open establishments that responded to the occupancy surveys requested by the NIS, there were 315 accommodations offering a total of 8578 bed places in 2021. In terms of the number of establishments, it is followed by the tourist territory of Sierra de Gata, Las Hurdes and Valle del Alagón, to the west of Valle del Jerte-La Vera, with 224 accommodations, and the city of Cáceres, with 216; moreover, Badajoz, the main city of Extremadura, only offers 27 accommodations. In terms of bed places, it is followed by Cáceres, which has less than half as many as Valle del Jerte-La Vera, with a total of 4146 bed places, and Valle del Ambroz–Tierras de Granadilla, with 3908 bed places. At the bottom of the list is Geoparque Villuercas-Ibores-Jara, with almost 1390 places.

With regard to its evolution during the four years of study, in Valle del Jerte-La Vera, both the number of accommodation establishments and the number of bed places offered have increased (Figure 9). In the case of the number of tourist accommodation establishments, hotel establishments have maintained their numbers throughout the period; equally, there has been a strong growth in rural accommodations, from 193 in 2018 to 209 in 2021, followed by non-hotel accommodations, which has increased progressively from 2018, with a total of 48, to 2021 with 59. This is due to the opening of 17 tourist flats in the area, as hostels and campsites have maintained their numbers. In relation to the number of vacancies offered by these tourist accommodations, the largest number is offered by non-hotel establishments (444)—specifically, tourist flats; this is followed by rural accommodations (244), while the number of vacancies in hotel establishments remains the same.

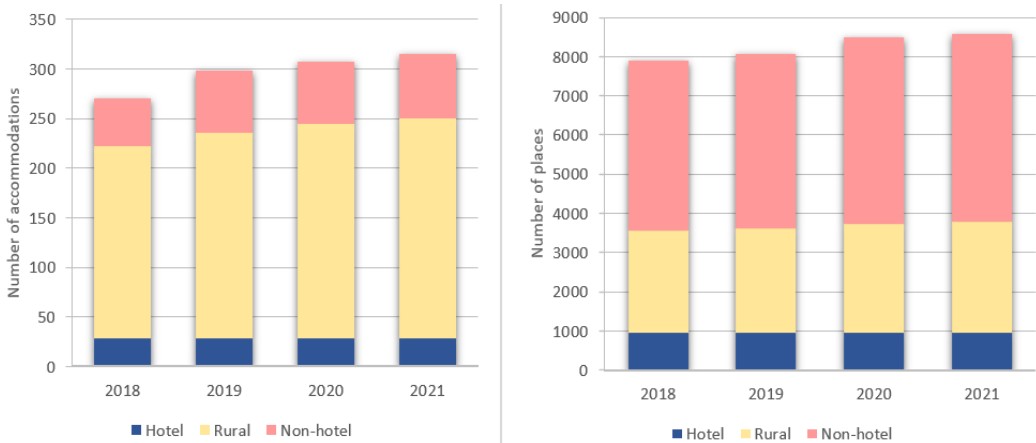

**Figure 9.** Graphs of the distribution of the number of accommodations and bed places offered by typology.

It can be seen how rural accommodations take precedence, being, in many municipalities, the only existing typology (Figure 10). This predominance is practically generalized in all of the municipalities of Valle del Jerte-La Vera, although, in municipalities, such as Talaveruela de la Vera, the number of rural accommodation establishments is equal to the number of hotels. In terms of the total number of establishments, the largest number is concentrated in Navaconcejo (34), followed by Jarandilla de la Vera (33); moreover, the majority of municipalities, i.e., 18, have fewer than 10 tourist establishments.

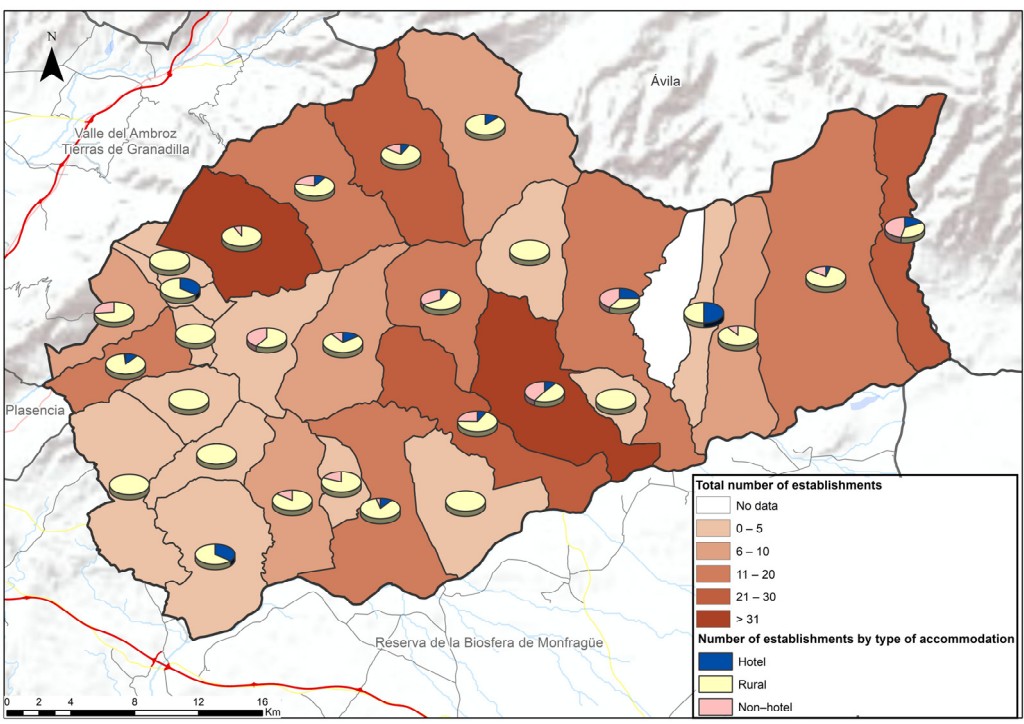

**Figure 10.** Map of the number of accommodations and their typologies by municipality.

With regard to the number of bed places offered (Figure 11), although the largest number of bed places is of the non-hotel type, by municipality, rural-type bed places predominate. Thus, in 21 municipalities, there is a greater number of bed places offered in rural accommodations. In addition, Valdastillas is particularly noteworthy, with a predominance of hotel accommodations. Overall, the largest number of bed places is con-

centrated in Jarandilla de la Vera, with 1656, followed by Losar de la Vera with 1101, while 10 municipalities in this area offer less than 50 bed places in their tourist accommodations.

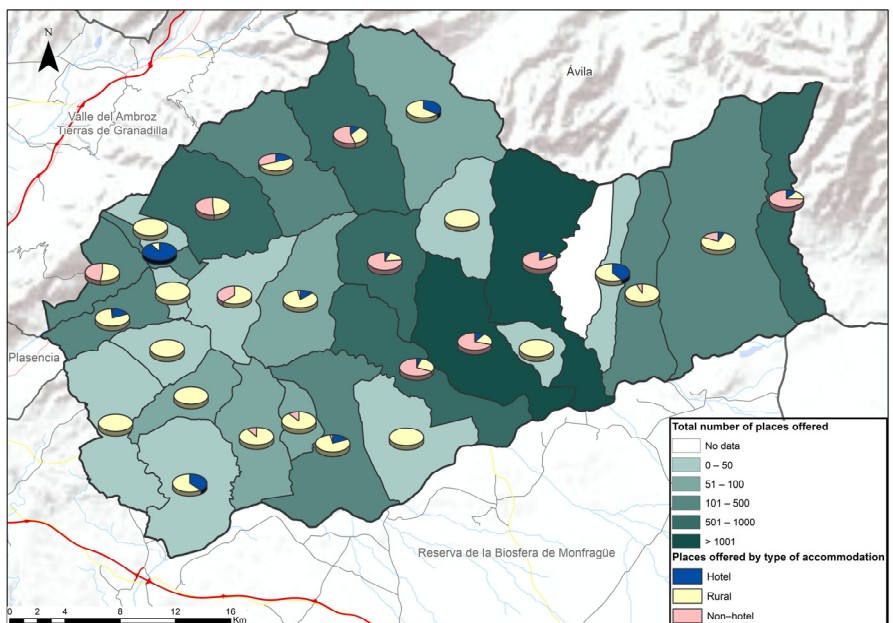

**Figure 11.** Map of the number of vacancies offered and their typologies by municipality.

Taking into account the data referring to the supply of tourist infrastructures in the tourist territory of Valle del Jerte-La Vera, specifically the number of establishments classified in the Register of Tourist Enterprises of the Directorate General of Tourism of the Regional Government of Extremadura, there were a total of 450 accommodations, of which 315 issued their responses to the occupancy survey to the NIS in the year 2021. This reflects the high degree of involvement of tourism entrepreneurs working in this territory. In this tourist territory, there is a high volume of rural accommodations (309), followed by non-hotel accommodations (97), and a small amount of hotel accommodations, which do not exceed 45 establishments. Analyzing the type of owner or proprietor of the tourist accommodations (Table 1), it can be seen that there is a practically an equal distribution between men (150), women (145), business groups, institutions and organizations (155), the latter being slightly higher. In non-hotel accommodations, there are more women than men serving as owners of the establishments; however, in the rest of the typologies, the number is lower. Closely related to the high number of women owners of tourist establishments is the support and funding of the LEADER and PRODER initiatives for women in rural areas, providing them with the opportunity to obtain employment and supplementary income [52], so that tourism has become a dynamic activity in these areas.

**Table 1.** Establishment owner and types of accommodations.

|  | **Hotel** | **Rural** | **Non-Hotel** | **Total** |
|---|---|---|---|---|
| Business group, institution or organization | 24 | 95 | 36 | 155 |
| Man | 12 | 112 | 26 | 150 |
| Woman | 8 | 102 | 35 | 145 |
| Total | 44 | 309 | 97 | 450 |

The trend towards the greater participation of women in this area is closely related to the existence of an agricultural economy linked to smallholdings with a lower productivity; in many cases, this generates the need for the greater diversification of family economies. In addition, the existence of a large number of natural and cultural resources in this tourist

area serves as a focus of attraction for travelers, and these are the key elements for the promotion and maintenance of accommodations.

In the study area, there is a significant representation of women in the management of tourist accommodations; however, in many cases, although a man is established as the owner, a woman is in charge of the management and development of the business. At this point, nowadays, female labor in rural areas is considered necessary to fix the population and generate new jobs, thus increasing the activity rates, and, in general, for the development of new economic activities that promote the process of productive diversification to effectively promote new development strategies in rural areas [53–55].

In terms of tourist demand, Valle del Jerte-La Vera received 121,403 travelers in the last year (2021), of which 90.7% (110,076) were nationals and 9.3% (11,328) internationals [56], with a homogeneous distribution, although with peaks at Easter (March–April) and in the summer months (June, July and August). Although with a lower volume of travelers, this trend follows the patterns of previous years, with the exception of 2020 due to the restrictions and effects of the pandemic crisis, which meant a decrease of more than 53% compared to 2019. Comparing the number of travelers that Valle del Jerte-La Vera receives with the rest of the tourist territories of Extremadura, over the last four years, it has established itself as the territory that receives the fourth-highest number of travelers, only surpassed by the main cities of the region, Cáceres, Mérida and Badajoz. Furthermore, in 2020, the territory under study received 9.6% of the total number of travelers in Extremadura, which reflects the fact that nature and rural destinations resisted, within their possibilities, the effects of the health crisis.

As can be seen in the following graph (Figure 12), the months with the lowest number of visitors coincide with the winter season (5.8%), while in spring (22.2%) and autumn (18.3%), they increase due to various factors, such as the good weather conditions and the presence of bank holidays or the Easter week, although the month with the highest number of visitors is August, with 23.1% of the annual total, which is why the period in which this territory receives the highest number of visitors is summer, reaching 53.7% in the last year. Thus, one of the problems in this area is the accentuated seasonality, where, in the summer months, the volume of visitors is 10 times higher than those received in January [36].

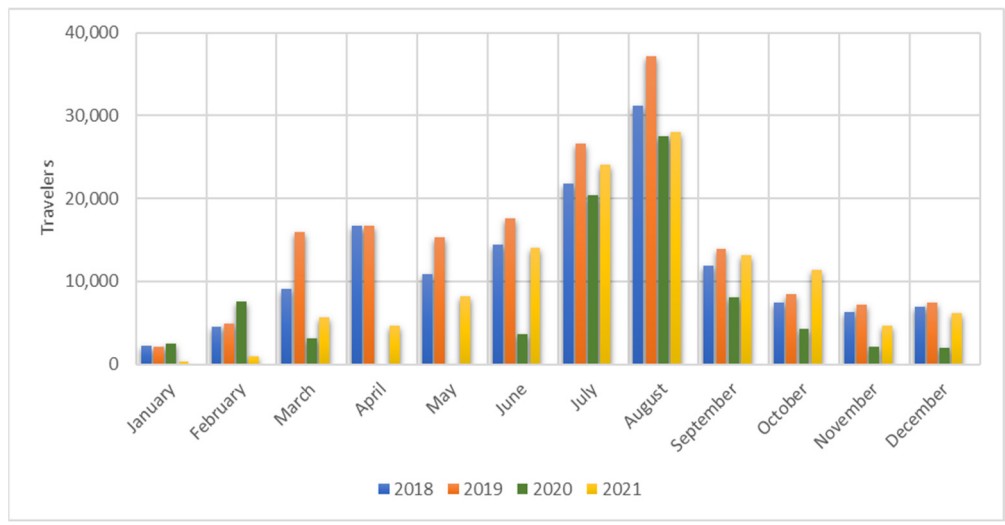

**Figure 12.** Graph of the evolution of total travelers during 2018, 2019, 2020 and 2021.

A more detailed analysis of the number of travelers shows a predominance of national travelers, who have accounted for over 90% of the total in the last four years. A homogeneous distribution pattern of national travelers is established throughout the year, with an upturn during the summer months, especially in August, when it reaches its peak. Mostly, this is due to the "paisano tourism", where people from the town who live in the cities

return to the town during the summer vacations. In the years prior to the pandemic, there was a peak in the months of Easter (March–April) and the Cherry Blossom Festival.

Moreover, the distribution of international travelers does not follow the same pattern as that of national travelers, which is distributed irregularly throughout the year (Figure 13), finding that the maximum value of these travelers also occurs in the summer season (June, July and August). It should be noted that, analyzing the last four years, the maximum number of international travelers was reached in 2021, with values of over 2000 travelers.

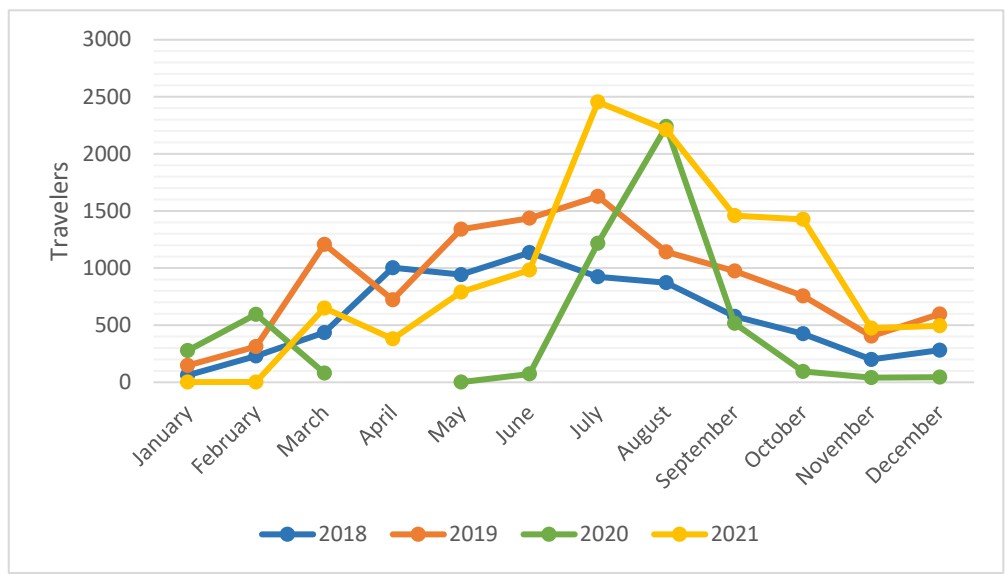

**Figure 13.** Graph of the evolution of international travelers for the years 2018, 2019, 2020 and 2021.

Analyzing travelers according to the accommodation in which they stay overnight, hotel establishments receive the most travelers (56,345 in 2019 and 24,602 in 2020), followed by rural accommodations (52,960 in 2019 and 22,395 in 2020) and campsites (42,523 in 2019 and 21,578 in 2020). There were lower values for tourist flats, which received almost 14,500 travelers in 2019 and slightly over 10,000 travelers during the pandemic. Finally, in the latter year, hostels did not even reach half of the number of travelers received in 2019.

In terms of overnight stays, these follow a similar pattern to the number of travelers. This is why the trend during the first two years of the study was an increase in these overnight stays, since the values increased from 368,348 overnight stays to 429,124, but, due to the crisis suffered, this trend was terminated. This is why, due to the reduction in the number of travelers, overnight stays experienced this decrease, which was even more accentuated as the values did not reach 218,000 overnight stays in 2020, a decrease of more than 49% with respect to 2019. It should be noted that in the last year, these overnight stays have been recovering, amounting to a total of 310,817 overnight stays in 2021. The upturn in overnight stays takes place in the summer months, with August being the month with the highest number of overnight stays. It should be noted that, in October last year, the number of overnight stays in 2018 and 2019 was exceeded, with an increase of more than 7000 overnight stays.

According to the distribution by the types of accommodations of these travelers in 2019, campsites are the establishments with the highest number of overnight stays in this tourist territory, with more than 132,000, while, in 2020, they did not reach 75,000 overnight stays. This was followed by rural accommodations, with 122,025 overnight stays, and hotel accommodations, with 117,556 overnight stays in 2019, a reduction of more than 50% in the year of the pandemic. Meanwhile, tourist flats received 37,000 overnight stays and hostels almost 20,000 overnight stays in 2019, values that dropped to 29,000 and 5600 overnight stays, respectively, in 2020.

## 4. Discussion and Conclusions

Extremadura is one of the communities with the highest increase in the rates of travelers and overnight stays, but these values are much lower than the national values. Bearing this in mind, the development of tourism is not the same in all the tourist areas of Extremadura, nor even between the northern areas of the region, due to the demographic, social and economic structures and the characteristics of each one. In this way, and according to this classification, there are territories with a high degree of tourist development in the cities of Cáceres, Mérida and Badajoz, followed by the study area, Valle del Jerte-La Vera. This area is characterized by a wide range of non-hotel and rural accommodations due to its great cultural, natural and scenic attractions. Furthermore, Valle del Jerte-La Vera has carried out an important tourist campaign, not only at a regional level, but also at national and even international levels, to establish an integrated offer combining its products par excellence, cherry and paprika, with its rich natural landscape and its gastronomy [40,57], which, together with its proximity to Madrid, has helped it to gain a foothold in both the national and international markets.

Based on the results obtained, it is verified that the demographic, economic and territorial characteristics are related to the development of tourism in various areas of the Extremadura region.

Firstly, and bearing in mind that the largest number of travelers and overnight stays in the region are concentrated in the cities of Cáceres, Mérida and Badajoz, focusing only on the tourist areas, Valle del Jerte and La Vera is the area that receives the most travelers, reaching more than 120,000 travelers last year, due to the impressive attraction of the Cherry Blossom show, in which visitors can admire the modeling of the terraces of the valley to adapt to the cultivation of cherries and the great mantle of white flowers that covers it, conveying the arrival of spring, where the PDO Jerte cherry will have special relevance and impact. Moreover, La Vera, similarly to Valle del Jerte, has numerous gorges and pylons that welcome travelers, especially in the summer months, as well as the local cuisine, which highlights the paprika PDO La Vera.

In both areas, rural tourism, agrotourism and nature tourism predominate, given the conditions of the area. In terms of distribution throughout the year, these travelers, who are mainly nationals, are established homogeneously throughout the year, with peaks at Easter and during the summer months, following the same pattern as the number of travelers to the region.

With regard to the supply, in terms of the number of accommodation establishments, the rural-type establishments stand out, with around 200 establishments, while the largest number of places available is concentrated in non-hotel accommodations, which has gradually increased to over 440 places at present, due to the proliferation of tourist flats in the area. It should be pointed out that, in their distribution by municipality, the county seats concentrate the greatest amount of accommodations and, likewise, practically all of the services available in this territory.

In relation to these variables, authors, such as Hernández-Mogollón, et al. [22] have established that visitors to this area of the region are predominantly hikers (86%), and they do not stay overnight in this territory; among the travelers who stay overnight in the area, it is mostly for a short period of time (one or two nights). With regard to the motivation for traveling, authors, such as Sánchez-Rivero et al. [36] state that the main reason for traveling to this area is the cultural visits, followed by rural tourism and river or gorge tourism. Moreover, in the analyses of other authors [42], the reason for such trips is the quality of the landscape (90.3%), the historical-artistic heritage (28.5%) or the quality of the typical products and gastronomy (22.6%). Furthermore, this type of tourist has been described as a person of mature age, with a medium-high level of purchasing power and with experience in tourism.

In this territory, tourism has become one of the activities with the greatest potential for endogenous development. Taking into account this influx of travelers and their seasonal nature, as mentioned above, numerous initiatives have been launched to promote and

boost the area with cultural, gastronomic, leisure and sporting activities, and even popular festivals, such as the Cherry Blossom and Jarramplas festivals. In terms of gastronomy, different resources of the area have been promoted, many of them with the quality seal of Protected Designations of Origin (PDO), such as the cherry of Valle del Jerte and paprika from La Vera, through gastronomic events, tapas fairs, etc. In the case of leisure and sporting activities, Agorreta et al. [58] state that, in recent years, numerous companies have been set up, linked to active tourism activities, guided routes, routes linked to water, canoeing, 4 × 4, etc., such as the Garganta de los Infiernos Mountain Race in Jerte (in April) and the Picota Bike Race Cycling Marathon in Navaconcejo (in October), among others. All of this is due to the fact that accommodations and restaurants need activities to complement this tourist offer.

In recent years, this leisure offer has been increasing with professional, competitive and quality companies, largely linked to active nature tourism, so it is necessary to promote other segments that complement this tourism, such as health, relaxation, gastronomy, etc. Nowadays, tourism marketing determines the promotion of the destination, which is why it is necessary to plan the tourist offer in order to promote the image of the area, which increases its value and positions it as a destination that tends to be more in demand and competitive.

Several authors [59–61] have analyzed the evolution and influence of tourism strategies in rural areas of our country, establishing at first that rural tourism served as a complement to agricultural incomes [14,19,23,62], but subsequently, after the economic crisis, this activity began to develop as the main activity capable of generating profits in many areas of the interior of the peninsula, as has been the case in Valle del Jerte-La Vera.

Due to the territorial characteristics of the areas in which it is developed, rural tourism is currently considered a territorial strategy for its contribution to the creation of employment and, consequently, to the fixation of the local population, due to its capacity to generate complementary income and, furthermore, to be capable of generating support and consolidation for the rural environment [18,63]. In this way, the protected areas are being used as a tourist resource, and the activities developed around them are generating a series of benefits, since, in areas such as Valle del Jerte-La Vera, they receive a large number of travelers and overnight stays, due, above all, to their proximity to Madrid, which is the main source of rural tourism travelers in Extremadura. Accessibility to the Spanish capital is a territorial variable and a determining factor in rural tourism [64,65].

This research has served to determine how the different variables that make up a territory (for example, the types of accommodations, the supply of natural resources) complement the tourist offer and to know whether they are positively or negatively influencing the attraction of travelers. For these reasons, quality tourism should continue to be promoted, especially in those areas where tourism cannot yet be considered as an engine of development and can be established as an activity that generates complementary income to the traditional agricultural one. Moreover, it is necessary to have the involvement of the public administrations and the private sector in new investments in the sector and de-seasonalize the demand through the promotion of new and more experiential products that contribute to an increase in the stay and income. Moreover, marketing strategies must continue to be developed to boost and promote the area, so that destinations such as Valle del Jerte-La Vera, attract a greater number of visitors and position themselves as prime destinations. All of this must be supported through planned and organized projects, with the collaboration and cooperation between the regions to promote the territory. Additionally, in this case, the inhabitants of Valle del Jerte-La Vera must commit themselves to the maintenance and conservation of their heritage and resources, because these are the main incentives for most of the tourists of this territory, particularly, and of Extremadura, in general.

Finally, the impact of the COVID-19 pandemic has constituted a turning point in tourism activities and this must serve as a lesson for the future, so as to be able to anticipate other crises, in order to lessen their effects [66]. Its impact has varied depending on

the territory in terms of its dependence on the tourism sector, the tourism model that it has developed and the traveler profile in which it has specialized [67]. This is why rural and mountain destinations, based on natural and cultural resources are benefiting from the increase in demand [68,69], as these territories can offer a product with the main characteristics in demand, related to low mass tourism, such as individual accommodations, open and wide spaces, territorial quality and services on offer. Even so, the tourism sector must be improved to ensure that this activity in the post-COVID-19 era is based on quality, innovation, sustainability and safety.

**Author Contributions:** Conceptualization, N.R.R., G.C.A., A.N.M. and F.L.B.; methodology, N.R.R., G.C.A. and F.L.B.; software, N.R.R., G.C.A. and A.N.M.; validation, N.R.R. and G.C.A.; formal analysis, N.R.R.; investigation, N.R.R.; resources, N.R.R., G.C.A., A.N.M. and F.L.B.; writing—original draft preparation, N.R.R. and F.L.B.; writing—review and editing, N.R.R., G.C.A. and A.N.M.; visualization, N.R.R. and G.C.A.; funding acquisition, N.R.R., G.C.A., A.N.M. and F.L.B. All authors have read and agreed to the published version of the manuscript.

**Funding:** This publication has been made possible thanks to funding granted by the Consejería de Economía, Ciencia y Agenda Digital de la Junta de Extremadura and by the European Regional Development Fund of the European Union through the reference grant GR21164 and with funding from the Ministry of Universities of the Government of Spain with the aid for university teacher training (FPU), reference FPU20/03830.

**Institutional Review Board Statement:** Not applicable.

**Informed Consent Statement:** Not applicable.

**Data Availability Statement:** The data presented in this study are available on request from the first author or corresponding author.

**Conflicts of Interest:** The authors declare no conflict of interest.

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
