# Peer review of "The Territory of Valle del Jerte-La Vera and Its Tourist Development (Extremadura, SW Spain)"

_land, doi:10.3390/land11122171_

Round 1

Reviewer 1 Report

The paper proposes an interesting study case about natural and cultural resources underlying the tourism potential of Valle del Jerte-La Vera area, in autonomous region of Extremadura, proposing an articulated cartography in order to demonstrate the close relationship which links both tourism supply and demand and the resources of area.

Although the aim is cleary descriptive, the paper shows some weaknesses related to theoretical and methodological approach.

Concerning the former, , it would be advisable to expand paragrapgh 1 with some insights on the scientific debate on the critical aspects of sustainable rural development and territorial regeneration processes in inner areas, affected by depopulation and ageing, centred on tourism diversification of agriculture, a phenomenon which would seem to emerge from the analysis proposed in the study case.

Therefore, some readings are recommended:

  • Maroto-Martos, J.C.; Voth, A.; Pinos-Navarrete, A. The Importance of Tourism in Rural Development in Spain and Germany. In Neoendogenous Development in European Rural Areas. Results and Lessons; Cejudo, E., Navarro, F., Eds.; Springer: Cham, Switzerland, 2020; pp. 181–205.
  • Saxena, G.; Clark, G.; Oliver, T.; Ilbery, B. Conceptualizing integrated rural tourism. Tour. Geogr. 2007, 9, 347–370.
  • Courtney, P.; Hill, G.; Roberts, D. The role of natural heritage in rural development: An analysis of economic linkages in Scotland. J. Rural Stud. 2006, 22, 469–484.
  • Hernández-Mogollón, J.; Campón-Cerro, A.; Leco-Berrocal, F.; Pérez-Díaz, A. Agricultural diversification and the sustainability of agricultural systems: Possibilites for the development of agrotourism. Environ. Eng. Manag. J. 2011, 10, 1911–1921.
  • Ohe, Y.; Kurihara, S. Evaluating the complementary relationship between local brand farm products and rural tourism: Evidence from Japan. Tour. Manag. 2013, 35, 278–283.
  • Sims, R. Food, place and authenticity: Local food and the sustainable tourism experience. J. Sustain. Tour. 2009, 17, 321–336.
  • Belliggiano, A., Garcia, E. C., Labianca, M., Valverde, F. N., & De Rubertis, S. (2020). The “eco-effectiveness” of agritourism dynamics in Italy and Spain: A tool for evaluating regional sustainability. Sustainability, 12(17), 7080.

From a methodological point of view, it would be advisable to expand the historical series, taking into consideration at least a decade, in order to better understand current trends and interpret their causes, revealing more clearly  any interference caused by the recent pandemic crisis. The category of agritourism should also be made explicit in the list of rural accommodation categories given in brackets in line 202, possibly stating the reasons for their exclusion.

Finally, it would be appropriate to outline future research developments in the concluding paragraph.

Minor revisions:

Line 105 - make author's name explicit

Line 215 - dissolve acronym and put it in brackets

Author Response

Dear Reviewer,
We would like to commence by thanking for your valuable time and constructive comments. Your expert knowledge of the field has helped us to strengthen the manuscript significantly. According to the valuable suggestions provided by you, we have revised the manuscript. We have done it highlighting the changed and using the Microsoft Word change control. At that point, we expose our comments.
Sincerely,
The authors

Reviewers
- In the Introduction section, the lines in which the structure of the article are presented have been modified (lines 74-77).
In the following, taking into account the above considerations, the literature review section is presented. Following this, the methodology used for the development of the re-search is established, as well as the delimitation of the study area…)
- Also, taking into account the literature review, a subsection has been established (line 80), extending this research by deepening the thematic studied and emphasizing the marketing and promotion of tourism (lines 81 - 116).
- In the Materials and Methods, the name of the author has been made explicit (line 129). In the same section the acronym has been dissolved and corrected (line 240).
- In addition, the category of agritourism has been explicitly included (line 227) since it is part of the classification of accommodations established by the National Institute of Statistics of Extremadura. It has been added:
and other buildings dedicated to the agritourism sector located in rural areas.
- It should be mentioned that it has only been possible to carry out the study since 2018 due to the fact that the methodology used for the extraction of the data by the Tourism Observatory of Extremadura has changed and, in addition, these are the data that are available. In future analyses this research will be expanded as more data becomes available.
- In the Discussion and Conclusions several bibliographical references have been included (lines 557-650), once the ones suggested by the reviewers have been read, and this section has been completed taking into account the analysis carried out and the implications of the research to know its contribution (lines 634 - 651).

Reviewer 2 Report

Authors’ study is to “analyze the tourism sector in the tourist territory of Valle del Jerte–La Vera, to obtain information about tourism activity from 2018 to the present day and its relationship with socioeconomic, demographic and heritage variables”. Below are suggestions to further improve the quality of the current paper.

1. Problem statement is not clearly presented in the introduction section. Research gap and research value should be clearly described to the extent that the paper generates meaningful implications.

2. The literature review section is missing. Authors should include and present the literature on tourism development, marketing, and promotion in the section of literature review.

3. Under the section of discussion, authors should clearly describe theoretical and managerial implications by adding the sub-section of theoretical implications and managerial implications, in which research value and contribution are clearly described. Generally, this paper has a lack of related literature review and insights into travel and tourism marketing and promotion.

Thus, authors are encouraged to address the issues by reviewing the related articles.Below are suggested articles authors should consider reviewing:

Chi, X., & Han, H. (2021). Emerging rural tourism in China’s current tourism industry and tourism behaviors: The case of Anji County. Journal of Travel and Tourism Marketing, 38(1), 58-74.

Pipatpong Fakfare, Jin-Soo Lee & Heesup Han (2022). Thailand tourism: a systematic review. Journal of Travel & Tourism Marketing, 39 (2), 188-214.

Author Response

(The authors gave the same response as above.)

Round 2

Reviewer 1 Report

Dear authors, the proposed revisions are acceptable. Congratulations for your work

Reviewer 2 Report

NA